# Potential impact of annual vaccination with reformulated COVID-19 vaccines: Lessons from the US COVID-19 scenario modeling hub

Sung-mok Jung[1], Sara L. Loo[2], Emily Howerton[3], Lucie Contamin[4], Claire P. Smith[2], Erica C. Carcelén[2], Katie Yan[3], Samantha J. Bents[5], John Levander[4], Jessi Espino[4], Joseph C. Lemaitre[1], Koji Sato[2], Clifton D. McKee[2], Alison L. Hill[2], Matteo Chinazzi[6], Jessica T. Davis[6], Kunpeng Mu[6], Alessandro Vespignani[6], Erik T. Rosenstrom[7], Sebastian A. Rodriguez-Cartes[7], Julie S. Ivy[7], Maria E. Mayorga[7], Julie L. Swann[7], Guido España[8], Sean Cavany[8], Sean M. Moore[8], T. Alex Perkins[8], Shi Chen[9], Rajib Paul[9], Daniel Janies[9], Jean-Claude Thill[9], Ajitesh Srivastava[10], Majd Al Aawar[10], Kaiming Bi[11], Shraddha Ramdas Bandekar[11], Anass Bouchnita[12], Spencer J. Fox[13], Lauren Ancel Meyers[11], Przemyslaw Porebski[14], Srini Venkatramanan[14], Aniruddha Adiga[14], Benjamin Hurt[14], Brian Klahn[14], Joseph Outten[14], Jiangzhuo Chen[14], Henning Mortveit[14], Amanda Wilson[14], Stefan Hoops[14], Parantapa Bhattacharya[14], Dustin Machi[14], Anil Vullikanti[14], Bryan Lewis[14], Madhav Marathe[14], Harry Hochheiser[4], Michael C. Runge[15], Katriona Shea[3], Shaun Truelove[2], Cécile Viboud[5], Justin Lessler[1,2]*

1 University of North Carolina at Chapel Hill, Chapel Hill, North Carolina, United States of America, 2 Johns Hopkins Bloomberg School of Public Health, Baltimore, Maryland, United States of America, 3 The Pennsylvania State University, University Park, Pennsylvania, United States of America, 4 University of Pittsburgh, Pittsburgh, Pennsylvania, United States of America, 5 Fogarty International Center, National Institutes of Health, Bethesda, Maryland, United States of America, 6 Northeastern University, Boston, Massachusetts, United States of America, 7 North Carolina State University, Raleigh, North Carolina, United States of America, 8 University of Notre Dame, Notre Dame, Indiana, United States of America, 9 University of North Carolina at Charlotte, Charlotte, North Carolina, United States of America, 10 University of Southern California, Los Angeles, California, United States of America, 11 University of Texas at Austin, Austin, Texas, United States of America, 12 University of Texas at El Paso, El Paso, Texas, United States of America, 13 University of Georgia, Athens, Georgia, United States of America, 14 University of Virginia, Charlottesville, Virginia, United States of America, 15 U.S. Geological Survey, Laurel, Maryland, United States of America

* jlessler@unc.edu

**Data Availability Statement:** All data are available at https://github.com/midas-network/covid19-scenario-modeling-hub/.

## Abstract

### Background

Coronavirus Disease 2019 (COVID-19) continues to cause significant hospitalizations and deaths in the United States. Its continued burden and the impact of annually reformulated vaccines remain unclear. Here, we present projections of COVID-19 hospitalizations and deaths in the United States for the next 2 years under 2 plausible assumptions about immune escape (20% per year and 50% per year) and 3 possible CDC recommendations for the use of annually reformulated vaccines (no recommendation, vaccination for those aged 65 years and over, vaccination for all eligible age groups based on FDA approval).

**Funding:** SJ, SLL, CPS, ECC, JCL, KS, CDM, ST, AH, and JL were supported by Centers for Disease Control and Prevention (200-2016-91781). CPS, ST, and AH were supported by the National Science Foundation (2127976). CPS, AH, ST, and JL were supported by the US Department of Health and Human Services; Department of Homeland Security; California Department of Public Health; Johns Hopkins University. JCL, CPS, AH, ST, and JL were supported by Amazon Web Services. JL (R01GM140564) and JCL (5R01AI102939) were supported by the National Institutes of Health. LC, JoL, JE, and HH were supported by NIGMS award 5U24GM132013. EH and KS were supported by NSF RAPID awards DEB-2028301, DEB-2037885, DEB-2126278, and DEB-2220903. KY was supported by NSF Grant No. DGE1255832. EH was supported by the Eberly College of Science Barbara McClintock Science Achievement Graduate Scholarship in Biology at the Pennsylvania State University. MC, JTD, KM, and AV were supported by HHS/CDC 6U01IP001137, HHS/CDC 5U01IP0001137, and the Cooperative Agreement no. NU38OT000297 from the Council of State and Territorial Epidemiologists (CSTE). ETR, JSI, MEM, and JLS were supported by TRACS/NIH grant UL1TR002489; CSTE and CDC cooperative agreement no. NU38OT000297. GE and SMM were supported by Scenario Modeling Hub Consortium fellowship. SMM was supported by NIAID R21AI164391. TAP was supported by NIGMS R35 MIRA program R35GM143029. AS and MAA were supported by NSF Awards 2135784 and 2223933, and Scenario Modeling Consortium Fellowship. KB, SRB, AB, SJF, and LAM were supported by CSTE NU38OT000297, CDC Supplement U01IP001136-Suppl, CDC 75D30122C14776 and NIH Supplement R01AI151176-Suppl. PP, SV, AA, BL, BK, JO, BH, HM AW, MM, JC, SH, PB, DM acknowledge support from SMC Fellowship 75D30121F00005-2005604290, VDH Grant PV-BII VDH COVID-19 Modeling Program VDH-21-501-0135, NSF Grant No. OAC-1916805, NSF Expeditions in Computing Grant CCF-1918656, DTRA subcontract/ARA S-D00189-15-TO-01-UVA, and UVA strategic funds. Model computation was supported by NSF ACCESS CIS230005 and UVA. The funders had no role in the design and conduct of the study; collection, management, analysis, and interpretation of the data; preparation, review, or approval of the manuscript; and decision to submit the manuscript for publication.

**Competing interests:** JE is president of General Biodefense LLC, a private consulting group for public health informatics, and has interest in READE.ai, a medical artificial intelligence solutions company. MR reports stock ownership in Becton

## Methods and findings

The COVID-19 Scenario Modeling Hub solicited projections of COVID-19 hospitalization and deaths between April 15, 2023 and April 15, 2025 under 6 scenarios representing the intersection of considered levels of immune escape and vaccination. Annually reformulated vaccines are assumed to be 65% effective against symptomatic infection with strains circulating on June 15 of each year and to become available on September 1. Age- and state-specific coverage in recommended groups was assumed to match that seen for the first (fall 2021) COVID-19 booster. State and national projections from 8 modeling teams were ensembled to produce projections for each scenario and expected reductions in disease outcomes due to vaccination over the projection period.

From April 15, 2023 to April 15, 2025, COVID-19 is projected to cause annual epidemics peaking November to January. In the most pessimistic scenario (high immune escape, no vaccination recommendation), we project 2.1 million (90% projection interval (PI) [1,438,000, 4,270,000]) hospitalizations and 209,000 (90% PI [139,000, 461,000]) deaths, exceeding pre-pandemic mortality of influenza and pneumonia. In high immune escape scenarios, vaccination of those aged 65+ results in 230,000 (95% confidence interval (CI) [104,000, 355,000]) fewer hospitalizations and 33,000 (95% CI [12,000, 54,000]) fewer deaths, while vaccination of all eligible individuals results in 431,000 (95% CI: 264,000–598,000) fewer hospitalizations and 49,000 (95% CI [29,000, 69,000]) fewer deaths.

## Conclusions

COVID-19 is projected to be a significant public health threat over the coming 2 years. Broad vaccination has the potential to substantially reduce the burden of this disease, saving tens of thousands of lives each year.

## Author summary

### Why was this study done?

- While Severe Acute Respiratory Syndrome Coronavirus 2 (SARS-CoV-2) is likely to pose a persistent threat to public health for the foreseeable future, regular revaccination with reformulated vaccines is considered a prominent mitigation tool.

- Questions exist regarding the effectiveness of annual vaccination campaigns and the optimal target age ranges, given the concentration of severe Coronavirus Disease 2019 (COVID-19) outcomes in older populations.

- The US COVID-19 Scenario Modeling Hub (SMH) has provided projections on the unfolding of the COVID-19 epidemic under various conditions, summarizing the results of multiple teams working on the same set of scenarios.

- Informed decisions on future vaccination policy need to be made with well-grounded projections of the likely course of COVID-19 epidemics and its impact under different vaccination scenarios.

Dickinson & Co., which manufactures medical equipment used in COVID-19 testing, vaccination, and treatment. JL has served as an expert witness on cases where the likely length of the pandemic was of issue. The remaining authors declare no competing interests.

**Abbreviations:** ACIP, Advisory Committee on Immunization Practices; CI, confidence interval; COVID-19, Coronavirus Disease 2019; FDA, Food and Drug Administration; LOP, linear opinion pool; PI, projection interval; SARS-CoV-2, Severe Acute Respiratory Syndrome Coronavirus 2; SMH, Scenario Modeling Hub.

## What did the researchers do and find?

- Applying the SMH approach, we projected the potential impact of COVID-19 from April 2023 to April 2025 and assessed the extent to which vaccination can reduce hospitalizations and deaths.

- Under plausible assumptions about viral evolution and waning immunity, COVID-19 will likely cause annual epidemics peaking in November to January over the two-year projection period.

- Though significant, hospitalizations and deaths are unlikely to reach levels seen in previous winters.

- The projected health impacts of COVID-19 are reduced by 10% to 20% through moderate use of reformulated vaccines.

## What do these findings mean?

- COVID-19 is projected to remain a significant public health threat in the coming years, exceeding the pre-pandemic mortality of influenza and pneumonia.

- Annual vaccination can reduce morbidity, mortality, and strain on health systems.

- While the projected impact of annual vaccination is significant, it is conditional on scenario assumptions including vaccine coverage and effectiveness.

## Introduction

Three and a half years after the Severe Acute Respiratory Syndrome Coronavirus 2 (SARS-CoV-2) virus first emerged in Wuhan, China, it seems the global community has transitioned from confronting Coronavirus Disease 2019 (COVID-19) as a pandemic emergency to managing it as an endemic, seasonally recurring virus [1]. While widespread immunity against SARS-CoV-2 has been achieved globally through vaccination and infections [2], the continued evolution of the virus causes antigenic changes and raises the potential for recurrent epidemics [3,4]. Current evidence suggests that both patterns of human contact and environmental factors contribute to seasonality in the intensity of SARS-CoV-2 transmission [5–7]. Combined, seasonality and ongoing "antigenic drift (i.e., gradual genetic changes in a virus evading prior population immunity [8])" of SARS-CoV-2 make it highly likely that the virus will pose a persistent threat to public health for the foreseeable future.

Going forward, one of the main tools for mitigating the impact of annual COVID-19 epidemics will be vaccination. As with influenza [9,10], continued antigenic drift of SARS-CoV-2 and intrinsic waning of the protection offered by previous vaccinations and infections (i.e., loss of immunity due to waning of immune protection, independent of the evolution of the virus) means regular revaccination with reformulated SARS-CoV-2 vaccines will be needed to mitigate the virus's impact [11]. However, legitimate questions exist about how effective annual vaccination campaigns can be, given SARS-CoV-2's rapid evolution, and what age ranges should be targeted, given the concentration of severe COVID-19 outcomes in older populations [12]. Hence, well-grounded projections of COVID-19's impact under different vaccination scenarios help inform future vaccination policy.

The US COVID-19 Scenario Modeling Hub (SMH) is a long-standing multi-team modeling effort that aims to project how the COVID-19 epidemic is likely to unfold in the mid- to long-term under various conditions [13,14]. These planning scenarios contrast various interventional strategies, characteristics of future viral variants, and other epidemiological or behavioral uncertainties, to provide projections of COVID-19 hospitalizations and deaths under each set of assumptions. By summarizing the results of multiple teams working on the same set of scenarios, the SMH takes advantage of the proven increased reliability of ensemble-based predictions over individual models [15]. Ensemble approaches have proven useful in multiple fields and across pathogens to inform public health policy, situational awareness, and individual decision-making [13].

Here, we present the results of applying the SMH approach to project the likely course of the COVID-19 epidemic in the United States over a two-year period (April 15, 2023 to April 15, 2025) under different assumptions about the average speed of antigenic drift and possible recommendations for the use of reformulated annual COVID-19 vaccines from the Centers for Disease Control and Prevention (CDC).

## Methods

To estimate the potential impact of vaccination on COVID-19 hospitalizations and deaths, we invited multiple teams in an open call to provide 2 years of projections for 6 scenarios within the SMH framework [14,15]. Teams had broad discretion in the details of model implementation within scenario definitions (see below). Individual team projections were combined to produce ensemble projections for each scenario as well as an ensemble estimate of the expected impact of vaccination.

### Scenario definitions

Six scenarios were created representing the intersection of 2 axes: one representing the average speed of antigenic drift (i.e., immune escape) over the two-year projection period, and the second representing differing assumptions about CDC recommendations for, and uptake of, a reformulated SARS-CoV-2 vaccine. The antigenic drift axis consisted of (1) a "low immune escape" scenario, where the SARS-CoV-2 virus evolves away from the immune signature of circulating variants at a rate of 20% per year (e.g., a vaccine with efficacy against symptomatic infection of 65% on June 15, 2023, is assumed to have an efficacy of $0.8 \times 0.65 = 52\%$ 1 year later in the absence of immune waning); and (2) a "high immune escape" scenario with an immune escape rate of 50% per year. The implementation of immune escape in their models was left at the discretion of teams (e.g., continuously or in stepwise occurrences; **S1 Table**) while ensuring that the annual levels align with the scenario definition.

The vaccination axis consisted of 3 levels based on possible COVID-19 vaccine recommendations under consideration by the CDC Advisory Committee on Immunization Practices (ACIP): (1) no recommendation for annual vaccination with a reformulated vaccine; (2) a recommendation for those aged 65 and above (65+); and (3) a recommendation for all ages eligible for vaccination based on the US Food and Drug Administration (FDA) approval [16]. Across all scenarios, the vaccine is assumed to be reformulated to match the predominant variants circulating as of June 15 each year and to become available to the public on September 1 of the same year. The annual uptake of reformulated vaccines in recommended groups is projected to follow the age group specific (0–17, 18–64, and 65+) uptake patterns observed for the first booster dose in each state (i.e., the first additional dose of vaccines after completing the primary series, authorized in September 2021) [17]. Uptake is assumed to saturate at levels reached 1 year after the recommendation (full uptake assumptions available on GitHub [18];

corresponding to 9% coverage in ages 0 to 17, 33% in 18 to 64, and 65% in 65+ nationally). Reformulated vaccines are presumed to have 65% vaccine effectiveness against symptomatic disease at the time of reformulation and immediately after receipt, with protection declining based on waning immunity and antigenic drift. This assumption was derived from a prior study showing a 60% vaccine effectiveness against emergency department encounters of the bivalent mRNA vaccine (fall 2022) [19], while considering potential underestimation due to immune waning and unreported previous SARS-CoV-2 antigen exposures. Vaccine effectiveness against severe outcomes was at the teams' discretion based on their best insights (**S1 Table**).

All contributing models were directed to incorporate waning immunity, with a requirement that the median waning time of protection against infection aligned with the designated range of 3 to 10 months. Furthermore, the incorporation of SARS-CoV-2 seasonality was required, though teams had discretion in terms of its implementation without any constraints on the timing and extent of seasonal forcing (e.g., not restricted to having a single seasonal peak; **S1 Table**). Teams were directed not to consider changes in non-pharmaceutical interventions over the projection period, given their limited implementation in 2023. Full scenario details are available on GitHub [18].

### Ensemble projections

Eight different modeling teams contributed projections of weekly incident and cumulative COVID-19 hospitalizations and deaths for April 15, 2023 to April 15, 2025 for all states and at the national level (1 additional team provided projections for only North Carolina based on their interest). Each team provided up to 100 representative epidemic trajectories for each scenario and outcome. Trajectories were used to generate a probability distribution of incident outcomes each week. Distributions at each week were combined using the trimmed-linear opinion pool method (LOP) to create ensemble projections (2 outermost values were trimmed while assigning equal weight to all remaining values) [15,20–22]. All reported numbers for incident and cumulative deaths and hospitalizations, and associated projection intervals (PIs), come from this ensemble.

To estimate the expected impact of vaccination, the mean and variance in cumulative deaths and hospitalizations were calculated over the period of interest based on submitted trajectories. Within each individual model, the expected impact of vaccination was determined by calculating the difference, or ratio, of projected deaths and hospitalizations between different vaccination scenarios sharing the same rate of immune escape, with variances estimated using the Delta method [23]. These individual model level estimates were then combined to produce an ensembled estimate of expected vaccine impact and associated confidence intervals (CIs) using standard meta-analysis techniques (with a random effects model) as implemented in the R package "*metafor*" [24,25]. We note that in estimating vaccine impact we (1) take the vaccine impacts estimated by each model and then ensemble those (rather than looking at the impact in ensemble estimates); and (2) use different techniques in combining vaccine impact estimates aimed at getting expected values and confidence intervals (rather than predictions intervals). Hence, vaccine impacts estimated from the meta-analysis are not directly reproducible by comparing ensemble projections for each scenario (which are not mathematically equivalent).

## Results

Based on the ensemble of projections from 8 contributing models under plausible assumptions about the viral evaluation and annual vaccination recommendations from the CDC, we project

National ensemble projections for COVID-19 hospitalizations

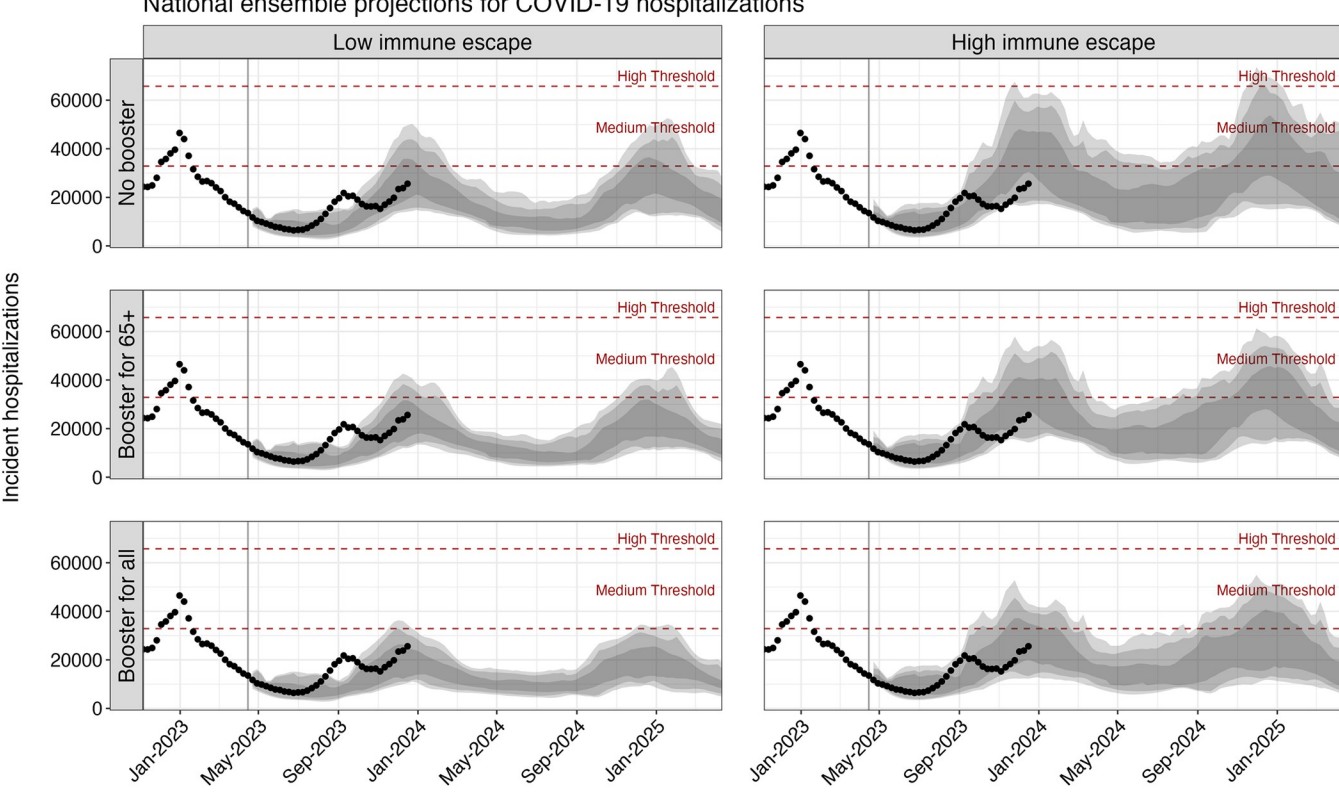

**Fig 1. Projected weekly COVID-19 hospitalizations in the United States across scenarios, April 2023–April 2025.** Ensemble projections from the COVID-19 SMH of national COVID-19 hospitalization for the period April 2023–April 2025 are shown by scenario. Dots indicate the observed weekly hospitalizations between December 1, 2022 and December 16, 2023. Shading from lightest to darkest represents 90%, 80%, and 50% projection intervals. Red dashed lines correspond to the CDC-designated COVID-19 community-level indicators: medium (10–19 weekly hospitalizations per 100,000) and high (>20 weekly hospitalizations per 100,000) levels. The vertical line on April 15, 2023, marks the start of the projection period. COVID-19, Coronavirus Disease 2019; SMH, Scenario Modeling Hub.

that between April 15, 2023 and April 15, 2025, the United States will experience annual COVID-19 epidemics peaking between November and January and causing approximately 1 million cumulative hospitalizations and 100,000 cumulative deaths each year (**Fig 1** and **Table 1**). The extent of COVID-19 impact over this period varies significantly by scenario, with 1.4 million (90% PI [983,000, 1,947,000]) hospitalizations and 130,000 (90% PI [71,000, 201,000]) deaths over the two-year projection period in the most optimistic scenario (reformulated vaccines recommended for all individuals, 20% immune escape) and 2.1 million (90% PI [1,438,000, 4,270,000]) hospitalizations and 209,000 (90% PI [139,000, 461,000]) deaths in the most pessimistic scenario (no recommendation, 50% immune escape) (**S1 Fig**). While significant, even in the most pessimistic scenario, we project deaths and hospitalizations are unlikely to be as high as the peak weekly hospitalizations seen in the first Omicron wave in early 2022 (150,000 hospitalizations per week). Furthermore, projected weekly hospitalizations are likely to remain at or below CDC-designated medium community transmission levels (10 to 19 weekly hospitalizations per year) [26] across all scenarios (**Fig 1**). There is moderate variation between states in peak timing and size of COVID-19 epidemic waves, although most generally follow national trends (**S2 and S3 Figs**).

Ensemble projections indicate that annual vaccination has the potential to substantially reduce both hospitalizations and deaths from COVID-19 (**Fig 2**). In high immune escape scenarios, if vaccination is confined to 65+, and uptake patterns mirror what was seen for the first booster

**Table 1. Projected national peak timing and peak size of hospitalizations across scenarios.**

| Scenario | April 15, 2023–April 14, 2024 | | | | April 15, 2024–April 15, 2025 | | | |
|---|---|---|---|---|---|---|---|---|
| | Peak timing | Peak size | Total hospitalizations | Total deaths | Peak timing | Peak size | Total hospitalizations | Total deaths |
| **High immune escape** | | | | | | | | |
| **No booster recommendation** | Dec 10 (Oct 15–Apr 14) | 42,000 (18,000–105,000) | 1,017,000 (767,000–2,058,000) | 100,000 (68,000–217,000) | Dec 15 (Oct 13–Apr 13) | 45,000 (17,000–90,000) | 1,093,000 (670,000–2,211,000) | 108,000 (71,000–244,000) |
| **Booster recommended for 65+** | Dec 10 (Oct 15–Feb 7) | 39,000 (17,000–91,000) | 943,000 (689,000–1,859,000) | 94,000 (55,000–178,000) | Dec 15 (Oct 13–Feb 23) | 41,000 (16,000–77,000) | 1,049,000 (584,000–1,959,000) | 99,000 (67,000–189,000) |
| **Booster recommended for all** | Dec 10 (Oct 8–Feb 18) | 35,000 (15,000–91,000) | 836,000 (595,000–1,723,000) | 82,000 (53,000–173,000) | Dec 8 (Jun 9–Feb 19) | 32,000 (14,000–77,000) | 949,000 (606,000–1,741,000) | 89,000 (64,000–182,000) |
| **Low immune escape** | | | | | | | | |
| **No booster recommendation** | Dec 13 (Aug 13–Apr 14) | 36,000 (16,000–81,000) | 825,000 (676,000–1,169,000) | 79,000 (57,000–124,000) | Dec 29 (Oct 27–Apr 13) | 35,000 (14,000–76,000) | 956,000 (578,000–1,304,000) | 85,000 (49,000–166,000) |
| **Booster recommended for 65+** | Dec 10 (Aug 13–Feb 18) | 34,000 (15,000–68,000) | 767,000 (620,000–1,020,000) | 70,000 (45,000–111,000) | Dec 22 (Oct 27–Mar 9) | 32,000 (13,000–65,000) | 857,000 (485,000–1,128,000) | 80,000 (34,000–109,000) |
| **Booster recommended for all** | Dec 3 (Apr 30–Mar 3) | 26,000 (13,000–57,000) | 670,000 (487,000–920,000) | 63,000 (38,000–101,000) | Dec 15 (Jun 12–Mar 9) | 28,000 (12,000–51,000) | 717,000 (496,000–1,027,000) | 67,000 (33,000–100,000) |

Each value represents the median projection with 90% PI below.

PI, projection interval.

dose, we would expect a reduction in hospitalizations of 8% (95% CI [5, 12]) compared to the no vaccination scenario and a reduction in deaths of 13% (95% CI [7, 18]). This corresponds to absolute reductions of 230,000 (95% CI [104,000, 355,000]) hospitalizations and 33,000 (95% CI [12,000, 54,000] deaths across the entire United States over the two-year projection period.

Expanding vaccination recommendations to all individuals would lead to substantial additional reductions in deaths and hospitalizations (**Fig 2**). Under the assumption that coverage equivalent to the first booster dose is attained, vaccination of all individuals reduces hospitalizations by 9% (95% CI [5, 13], *N* = 198,000, 95% CI [120,000, 276,000]) and deaths by 8% (95% CI [3, 14], *N* = 16,000, 95% CI [11,000, 22,000]) compared to vaccination of 65+ alone in high immune escape scenarios. This corresponds to a total reduction of 17% (95% CI [12, 22], *N* = 431,000, 95% CI [264,000, 598,000]) in hospitalizations and 20% (95% CI [12, 28], *N* = 49,000, 95% CI [29,000, 69,000]) in deaths compared to the no vaccination scenario. Results are similar in low immune escape scenarios.

A significant factor contributing to state-level variation in the projected impact of vaccine recommendations is the assumed uptake level of reformulated vaccines (**Figs 3, S4, and S5**). States with higher coverage among 65+ are anticipated to experience substantial reductions in hospitalizations, exceeding 150 per 100,000 in high immune escape scenarios, if the reformulated vaccines are recommended to all. In contrast, the state with the lowest coverage in 65+, North Carolina, is expected to witness reductions of less than 75 per 100,000.

## Discussion

Based on the ensemble of projections from 8 modeling teams for the next 2 years (April 2023 to April 2025), it is expected that COVID-19 will remain a persistent public health threat in the

Percent prevented (95% CI)　　　　Total prevented (95% CI)

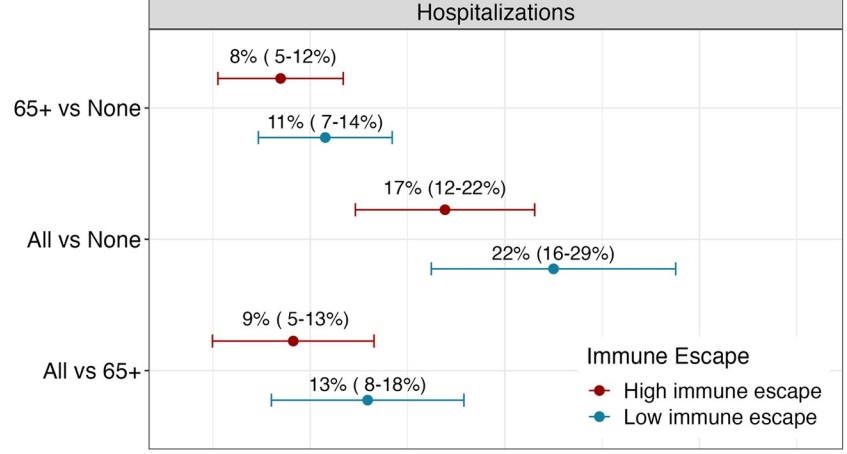
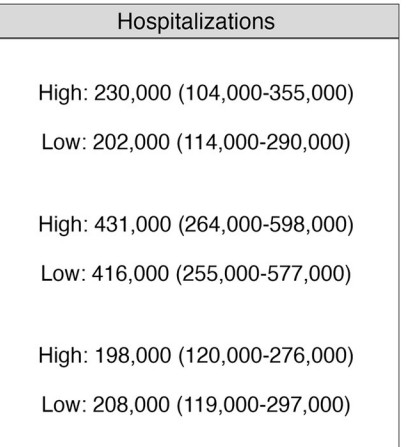

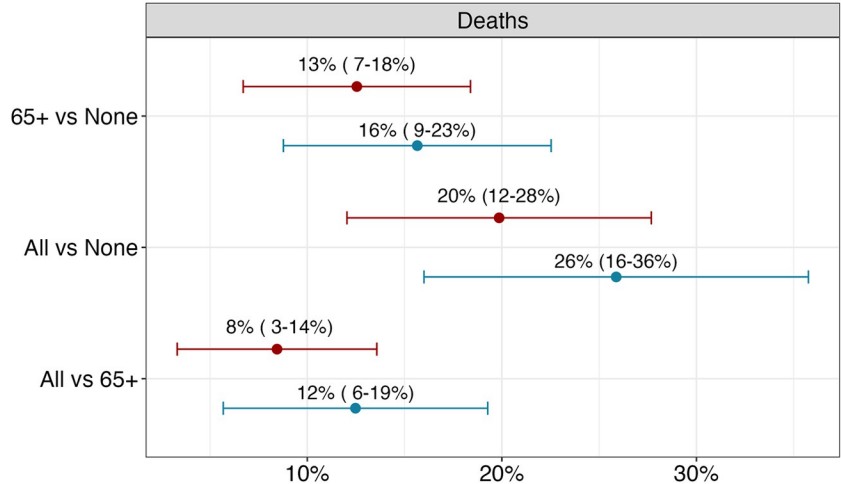
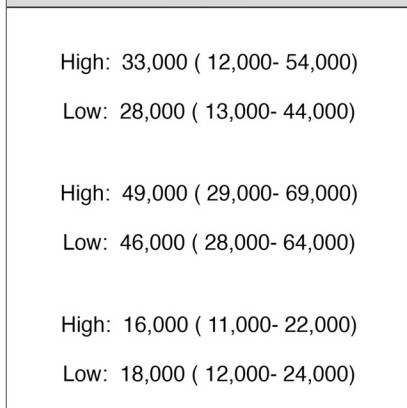

**Fig 2. Percent and total prevented COVID-19 hospitalizations and deaths by annual vaccination recommendation with reformulated vaccines.** Relative and absolute differences in cumulative hospitalizations and deaths over the next 2 years (April 2023–April 2025) between different vaccination recommendations. Red and blue dots and error bars represent the median and 95% CI of percent prevented outcomes in high and low immune escape scenarios (50% per year and 20% per year), respectively. CI, confidence interval; COVID-19, Coronavirus Disease 2019.

United States for the foreseeable future. Nevertheless, our projections highlight that annual vaccination with reformulated vaccines can substantially mitigate this burden if coverage reaches levels observed for the first (i.e., fall 2021) COVID-19 booster.

Across all scenarios, our projections indicate that COVID-19 hospitalizations and deaths would be substantially less than what was seen in the early stages of the pandemic (e.g., between April 2021 and April 2023, there were 4.2 million hospitalizations and 570,000 deaths [27]). Nonetheless, COVID-19 is projected to remain one of the leading causes of death in the United States [28]. For context, in our most pessimistic scenario (no CDC vaccine recommendation, high immune escape), annual COVID-19 mortality is expected to be similar to pre-pandemic mortality from Alzheimer's disease (**Fig 4**), while in the most optimistic scenario (vaccines recommended for all, low immune escape) mortality would be similar to that seen from diabetes in the pre-pandemic period. In all cases, COVID-19 mortality is projected to exceed that of influenza and pneumonia.

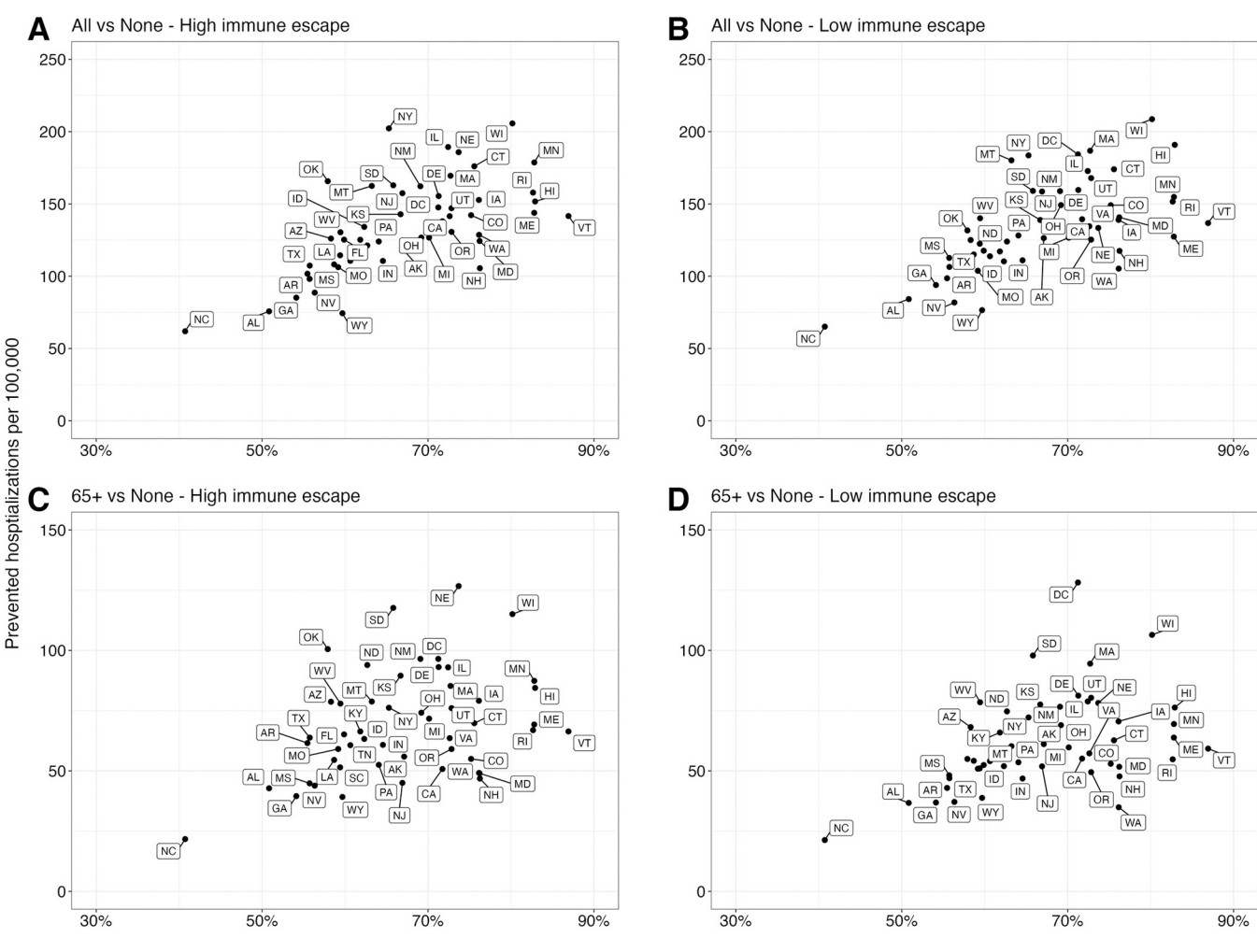

**Fig 3. Relationship between prevented COVID-19 hospitalizations and assumed vaccine coverage in individuals aged 65 and above across US states.** The relationship between the cumulative difference in COVID-19 hospitalizations for the next 2 years (April 2023–April 2025) under different vaccination recommendations and assumed vaccine uptake among those aged 65 and above (65+) in each US state: **(A and B)** vaccination of all compared to no vaccination and **(C and D)** vaccination of 65+, compared to no vaccination. The x-axis represents the assumed vaccine coverage among 65+ at saturation considering the higher severity in 65+ (likely to have the most significant contribution to decreasing hospitalizations). Dots in each panel correspond to individual US states. COVID-19, Coronavirus Disease 2019.

While the projected impact of annual vaccination on disease burden is significant, it is highly dependent on assumed vaccine uptake. This gives us reason for both caution and hope. Previous CDC booster recommendations, including that for the 2022 reformulated vaccine (i.e., bivalent vaccines authorized in August 2022), have not achieved the coverage observed for the first booster [29]. Reduced coverage would substantially blunt the impact of any vaccine recommendations. However, it is worth noting that many states where we assume low vaccination coverage, such as North Carolina and Pennsylvania, have not historically been ranked among the states with the lowest vaccine coverage for annual influenza vaccines [30], suggesting potential for increasing vaccine uptake in these regions.

Among 6 considered scenarios, the one with high immune escape (50% per year) and CDC vaccine recommendation for all age groups aligns most closely with real-world practices. The CDC advised vaccinating all individuals aged over 6 months on September 12, 2023 [31], and

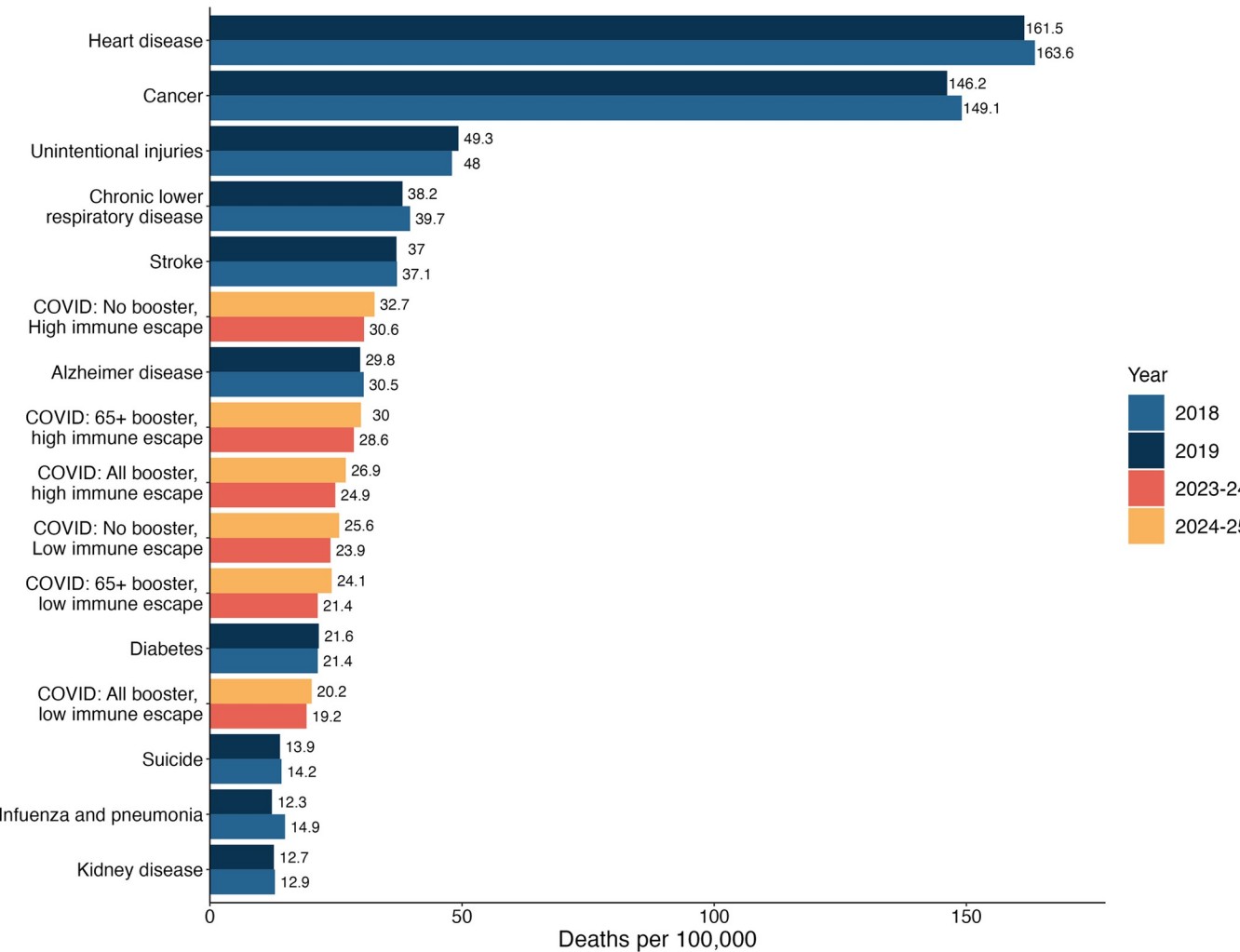

**Fig 4. Comparison between the projected COVID-19 mortality by scenario and the 10 leading causes of pre-pandemic mortality in the United States.**
Projected COVID-19 mortality by scenario and by period (April 2023–April 2024 and April 2024–April 2025) are compared with the 10 leading causes of mortality in the United States, which were obtained from the CDC age-adjusted disease burden rates in the pre-pandemic period [28]. COVID-19, Coronavirus Disease 2019.

the predominant variant in December 2023 (Omicron EG.5.1) was suggested to have around 17% immune escape compared to the preexisting variant in June 2023 (Omicron XBB.1.5). This is equivalent to an immune escape of around 40% per year, assuming the same transmissibility between 2 variants [32]. Our ensemble projections in this scenario appear to align well with the empirically observed national-level hospitalizations, yet some discrepancy was noted in September to October 2023, primarily attributed to faster resurgences in southern states [14]. In the discrepancy period, assumptions of vaccine coverage matched well with realized uptake, suggesting that factors other than vaccine assumptions drove the difference between observed and projected disease dynamics. We note that later in the fall of 2023, the observed reformulated vaccine uptake saturated at a lower level than our all-age scenario (although vaccine coverage observations are well bracketed by our set of scenarios; **S6 Fig**). However, state-level uptake patterns were comparable with the range of scenario assumptions in some states (**S7 Fig**) [33], particularly among 65+, who are likely to have significant contributions to reducing severe outcomes. Of note, our study primarily focuses on projecting the potential

advantages of annual vaccination (predicting the likely course of the epidemic given the scenario, rather than forecasting) to inform public health authorizations before the actual vaccination campaign begins; hence, our assumed uptake patterns in any individual scenario may not necessarily mirror the observed ones. Nevertheless, caution should be exercised when interpreting our projected hospitalizations and deaths averted by annual vaccination, as these outcomes are likely somewhat overestimated due to such discrepancies in vaccine uptake patterns. Additionally, in scenarios with vaccination recommendations to all individuals, the ensemble outperforms individual models, wherein most show either over- or underconfidence relative to the ensemble (**S8 Fig**). Such improvement of the ensemble over individual models aligns with our earlier findings based on prior rounds of SMH projections [15].

Our ensemble projections have potential implications for countries beyond the United States, where regular revaccination serves as a key strategy against COVID-19. In light of this global relevance, our projections provide insight into the benefits of annual vaccination in mitigating the disease burden, along with related work conducted in the European context [34]. However, it is essential to note that the magnitude of impacts may vary across countries due to differing epidemiological and demographic factors. In particular, variations in age distribution, circulating variables, transmission dynamics, and time-varying immunity within each age group can substantially influence the impact of annual vaccination efforts.

As with any attempt to project into the future, our projections come with major caveats and limitations. First and foremost, scenario projections are conditional on often strict scenario assumptions. Both vaccine coverage and effectiveness might deviate considerably from scenario assumptions, although historical trends of influenza vaccination suggest that achieving higher coverage is unlikely, especially in older populations [30]. Additionally, for simplicity, most teams assumed equivalent vaccine effectiveness against infection and symptomatic disease, potentially underestimating the vaccine impacts by neglecting protection against asymptomatic infections [35]. Furthermore, our scenarios did not consider interactions with other infectious diseases, but they may impact our projections if there are significant changes in risk perception or healthcare burden during the co-circulation of respiratory infectious diseases (e.g., tripledemic in the 2022 to 2023 season [36]). Nevertheless, projections of the combined impact of multiple pathogens for the 2023 to 2024 season suggest a probable lower impact on the healthcare system compared to the prior season [37]. Second, the potential impact resulting from variations in the details of the modeling approach (e.g., seasonality) and parameter values, determined at the teams' discretion, were not quantified due to the multi-team and real-time operational nature of the SMH framework. A hub structure is particularly useful when there is valid scientific uncertainty about the role of specific drivers of disease dynamics, including seasonality. Third, to accommodate diverse modeling approaches, we focused on aggregated projections of hospitalizations and deaths across all age groups for each scenario, while the scenarios were designed with different age-specific vaccine recommendations. Lastly, if future variants differ in intrinsic transmissibility or disease severity from that of the current Omicron lineages, the projected disease burden may alter accordingly. Furthermore, all scenarios were built on the assumption of continuous immune escape with a constant rate. However, the emergence of new SARS-CoV-2 variants showing a significant level of antigenic change within a very short span (e.g., Omicron [38,39]) could increase the disease burden far beyond these projections.

Despite its limitations, ensembling scenario-based projections from multiple teams has proven to be useful for estimating COVID-19's future burden and the potential benefits of vaccination, providing valuable information for public health planning [13,15]. Our results show that COVID-19 will likely remain a major threat to human health in the United States in the coming years. In the face of this threat, broad vaccination against SARS-CoV-2 has the potential to save tens of thousands of lives each year.

## Supporting information

**S1 Fig. Projected cumulative COVID-19 hospitalizations and deaths in the United States by scenario, April 2023–April 2025.** Ensemble projections for cumulative COVID-19 hospitalization and deaths in the United States for the next 2 years (April 2023–April 2025) are shown by scenario. Solid lines indicate the median of projected outcomes, and dash lines and shades indicate their 90% projection intervals. Each color represents different annual vaccination recommendations (no recommendation, reformulated vaccines recommended for those aged 65 and above, and recommended for all age groups). Dots indicate the observed cumulative hospitalizations and deaths from April 15, 2023 and December 16, 2023.
(TIF)

**S2 Fig. State-level peak COVID-19 hospitalizations in high immune escape scenarios by season and vaccination scenario.** The peak hospitalizations per 100,000 over the next 2 years (April 2023–April 2025) under high immune escape assumption are shown by US state and by vaccination scenario (no recommendation, reformulated vaccines recommended for those aged 65 and above, and recommended for all age groups). The color variation denotes the order of US states in the peak hospitalizations by scenario and season. Shades of yellow indicate states with lower values and shades of blue indicate states with higher values. For visualizations, square root scaling was applied in x-axes.
(TIF)

**S3 Fig. State-level peak timing of COVID-19 hospitalizations in high immune escape scenarios by season and vaccination scenario.** The peak timing of hospitalizations under high immune escape assumption is shown by US state and by vaccination scenario (no recommendation, reformulated vaccines recommended for those aged 65 and above, and recommended for all age groups). The color variation denotes the order of US states in the peak timing of COVID-19 hospitalizations by scenario and season. Shades of blue indicate states with an earlier peak and shades of yellow indicate states with a later peak.
(TIF)

**S4 Fig. State-level percent prevented COVID-19 hospitalizations between the annual vaccination scenarios from April 2023 to April 2025 by scenario.** Relative differences in cumulative COVID-19 hospitalizations over the next 2 years (April 2023–April 2025) between different vaccination scenarios are shown by immune escape level and by US state. The color variation denotes the order of US states in the percent prevented hospitalizations by scenario. Shades of yellow indicate states with lower values and shades of blue indicate states with higher values.
(TIF)

**S5 Fig. State-level percent prevented COVID-19 deaths between the annual vaccination scenarios from April 2023 to April 2025 by scenario.** Relative differences in cumulative COVID-19 deaths over the next 2 years (April 2023–April 2025) between different vaccination scenarios are shown by immune escape level and by US state. The color variation denotes the order of US states in the percent prevented deaths by scenario. Shades of yellow indicate states with lower values and shades of blue indicate states with higher values.
(TIF)

**S6 Fig. Comparison between the assumed and observed annual uptake of COVID-19 reformulated vaccines at the national level in the United States. (A)** Solid lines represent the assumed national-level annual uptake of reformulated vaccines by age group, projected to follow the uptake patterns for the first booster dose (authorized in September 2021). Dashed lines

indicate the empirically observed uptake as of February 24, 2024, sourced from the CDC website, covering Puerto Rico and the Virgin Islands which are not accounted for in the assumed national-level uptake. Each age group is represented by a different color. **(B)** Observed and assumed annual uptake of reformulated vaccines among individuals aged 18 and over at the national level. Each color represents a different vaccine coverage data.
(TIF)

**S7 Fig. Comparison between the assumed and observed annual uptake of COVID-19 reformulated vaccines by US state.** Solid lines represent the assumed state-level annual uptake of reformulated vaccines by age group, projected to follow the uptake patterns for the first booster dose (authorized in September 2021). Dots indicate the monthly observed uptake as of February 24, 2024, sourced from the CDC website. Each age group is represented by a different color.
(TIF)

**S8 Fig. Quantile-quantile (QQ) plot for assessing the performance of models regarding cumulative COVID-19 hospitalizations and deaths in the United States.** The actual coverage of each model, regarding cumulative hospitalizations and deaths as of December 16, 2023, is plotted against its expected coverage. Coverage measures the percentage of observations that fall within a given prediction interval (e.g., for a 90% prediction interval, expected coverage is 90%). Coverage was calculated across all locations and projection weeks. The dashed lines represent the expected relationship (expected coverage is equal to actual coverage), where a line below indicates models are overconfident (actual coverage is less than expected coverage), and above the line means models are underconfident (actual coverage is more than expected coverage). The black solid lines depict the ensemble model, while each colored line represents contributing individual models. Following the CDC recommendation for reformulated vaccines (published on September 12, 2023), only scenarios with vaccination recommendations to all individuals were included.
(TIF)

**S1 Table. Detailed description of individual models.**
(DOCX)

## Acknowledgments

JHU_IDD-flepiMoP: S. Jung, S. L. Loo, C. P. Smith, J. C. Lemaitre, K. Sato, C. D. McKee, A. L. Hill, S. Truelove, J. Lessler; MOBS-NEU-GLEAM_COVID: M. Chinazzi, J. T. Davis, K. Mu, A. Vespignani; NotreDame-FRED: G. España, S. Cavany, A. Perkins, S. M. Moore; UNCC-hierbin: S. Chen, R. Paul, D. Janies, J-C. Thill; USC-SIkJalpha: A. Srivastava, M. A. Aawar; UTA-ImmunoSERIS: K. Bi, S. R. Bandekar, A. Bouchnita, S. J. Fox, L. A. Meyers; UVA-adaptive/UVA-EpiHiper: P. Porebski, S Venkatramanan, J. Chen, A. Adiga, B. Klahn, B. Hurt, A. Wilson, S. Hoops, P. Bhattacharya, D. Machi, J. Outten, H. Mortveit, A. Vullikanti, B. Lewis, M. Marathe; NCSU-COVISM: E. R. Rosenstrom, S. A. Rodriguez-Cartes, J. S. Ivy, M. E. Mayorga, J. L. Swann; Coordination team: S. Jung, S. L. Loo, E. Howerton, L. Contamin, C. P. Smith, E. Carcelén, K. Yan, S. J. Bents, J. Espino, J. Levander, H. Hochheiser, M. C. Runge, K. Shea, S. Truelove, C, Viboud, and J. Lessler.

The findings and conclusions in this report are those of the authors and do not necessarily represent the official position of the US National Institutes of Health or Department of Health and Human Services. Any use of trade, firm, or product names is for descriptive purposes only and does not imply endorsement by the US Government.

## Author Contributions

**Conceptualization:** Harry Hochheiser, Michael C. Runge, Katriona Shea, Shaun Truelove, Cécile Viboud, Justin Lessler.

**Data curation:** Sung-mok Jung, Sara L. Loo, Emily Howerton, Lucie Contamin, Claire P. Smith, Erica C. Carcelén, Katie Yan, Samantha J. Bents, Joseph C. Lemaitre, Koji Sato, Clifton D. McKee, Alison L. Hill, Matteo Chinazzi, Jessica T. Davis, Kunpeng Mu, Alessandro Vespignani, Erik T. Rosenstrom, Sebastian A. Rodriguez-Cartes, Julie S. Ivy, Maria E. Mayorga, Julie L. Swann, Guido España, Sean Cavany, Sean M. Moore, T. Alex Perkins, Shi Chen, Rajib Paul, Daniel Janies, Jean-Claude Thill, Ajitesh Srivastava, Majd Al Aawar, Kaiming Bi, Shraddha Ramdas Bandekar, Anass Bouchnita, Spencer J. Fox, Lauren Ancel Meyers, Przemyslaw Porebski, Srini Venkatramanan, Aniruddha Adiga, Benjamin Hurt, Brian Klahn, Joseph Outten, Jiangzhuo Chen, Henning Mortveit, Amanda Wilson, Stefan Hoops, Parantapa Bhattacharya, Dustin Machi, Anil Vullikanti, Bryan Lewis, Madhav Marathe, Harry Hochheiser, Michael C. Runge, Katriona Shea, Shaun Truelove, Cécile Viboud, Justin Lessler.

**Formal analysis:** Sung-mok Jung, Sara L. Loo, Emily Howerton, Lucie Contamin, Claire P. Smith, Justin Lessler.

**Funding acquisition:** Alessandro Vespignani, Julie L. Swann, Sean M. Moore, Lauren Ancel Meyers, Bryan Lewis, Madhav Marathe, Harry Hochheiser, Katriona Shea, Shaun Truelove, Justin Lessler.

**Supervision:** Alessandro Vespignani, Julie L. Swann, Sean M. Moore, Shi Chen, Ajitesh Srivastava, Lauren Ancel Meyers, Bryan Lewis, Madhav Marathe, Harry Hochheiser, Michael C. Runge, Katriona Shea, Shaun Truelove, Cécile Viboud, Justin Lessler.

**Visualization:** Sung-mok Jung, Sara L. Loo, Emily Howerton.

**Writing – original draft:** Sung-mok Jung, Justin Lessler.

**Writing – review & editing:** Sung-mok Jung, Sara L. Loo, Emily Howerton, Lucie Contamin, Claire P. Smith, Erica C. Carcelén, Katie Yan, Samantha J. Bents, John Levander, Jessi Espino, Joseph C. Lemaitre, Koji Sato, Clifton D. McKee, Alison L. Hill, Matteo Chinazzi, Jessica T. Davis, Kunpeng Mu, Alessandro Vespignani, Erik T. Rosenstrom, Sebastian A. Rodriguez-Cartes, Julie S. Ivy, Maria E. Mayorga, Julie L. Swann, Guido España, Sean Cavany, Sean M. Moore, T. Alex Perkins, Shi Chen, Rajib Paul, Daniel Janies, Jean-Claude Thill, Ajitesh Srivastava, Majd Al Aawar, Kaiming Bi, Shraddha Ramdas Bandekar, Anass Bouchnita, Spencer J. Fox, Lauren Ancel Meyers, Przemyslaw Porebski, Srini Venkatramanan, Aniruddha Adiga, Benjamin Hurt, Brian Klahn, Joseph Outten, Jiangzhuo Chen, Henning Mortveit, Amanda Wilson, Stefan Hoops, Parantapa Bhattacharya, Dustin Machi, Anil Vullikanti, Bryan Lewis, Madhav Marathe, Harry Hochheiser, Michael C. Runge, Katriona Shea, Shaun Truelove, Cécile Viboud, Justin Lessler.

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
