## [Editor Report · Decision Letter 0]

2 Nov 2023

Dear Dr Lessler, 

Thank you for submitting your manuscript entitled "Potential impact of annual vaccination with reformulated COVID-19 vaccines: lessons from the U.S. COVID-19 Scenario Modeling Hub" for consideration by PLOS Medicine.

Your manuscript has now been evaluated by the PLOS Medicine editorial staff and I am writing to let you know that we would like to send your submission out for external peer review.

Please re-submit your manuscript within two working days, i.e. by Nov 06 2023 11:59PM.

Kind regards,

Katrien G. Janin, PhD

Senior Editor

PLOS Medicine

---

## [Decision Letter · Decision Letter 1]

4 Jan 2024

Dear Dr. Lessler,

Thank you very much for submitting your manuscript "Potential impact of annual vaccination with reformulated COVID-19 vaccines: lessons from the U.S. COVID-19 Scenario Modeling Hub" (PMEDICINE-D-23-03160R1) for consideration at PLOS Medicine. 

Your paper was evaluated and discussed among all the editors here. It was also discussed with an academic editor with relevant expertise, and sent to independent reviewers, including a statistical reviewer. The reviews are appended at the bottom of this email and any accompanying reviewer attachments can be seen via the link below:

[LINK]

In light of these reviews, we will not be able to accept the manuscript for publication in the journal in its current form, but we would like to consider a revised version that addresses the reviewers' and editors' comments. We cannot make any decision about publication until we have seen the revised manuscript and your response, and we plan to seek re-review by one or more of the reviewers. 

We expect to receive your revised manuscript by Jan 29 2024 11:59PM. Please email us (plosmedicine@plos.org) if you have any questions or concerns.

We look forward to receiving your revised manuscript. 

Sincerely,

Katrien Janin, PhD

PLOS Medicine

plosmedicine.org

Comments from the Academic Editor:

1. Please add line numbers to aid with reviewing.

2. Methods, Page 4: Clarify what the “… recommendation for eligible groups…” entails. I.e. which groups would be eligible for vaccination under this scenario

3. Methods/Results would benefit from a greater description (and comparison) of the models included in the ensemble (perhaps in the form of a table with supporting text?)

4. Seasonality: My understand from the limited description in the methods is that all modelling teams assumed some form of annual seasonality (presumably with a single winter peaks, given the predictions presented in Figure 1?). However, over 2023, many countries continued to see several peaks throughout the year, mostly unrelated to season. The authors should undertake sensitivity analysis, allowing greater flexibility in seasonality.

5. Major (perhaps unjustified assumption) is that annual booster vaccination uptake will mirror uptake when boosters became first available in Sept 2021. US data shows that more recent uptake has been considerably lower, and varying by age group. My feeling is that future rounds of booster uptake will continue to be low. Thus, I am not convinced that this assumption holds and gives an overly optimistic outcome. The authors should revise analysis to include modelling of scenarios where booster uptake is lower - and in line with more recent US estimates - and discussion in greater detail the implications of this.

6. Wasn’t clear to me whether any models included “boosting” effects of repeated infection over the study period. This needs to be clearly described, perhaps in the additional table suggested above.

7. Reader will want to see more detailed descriptions of the age-group specific effects under various scenarios (currently in a figure, but greater description in Results, and in Discussion section would be helpful)

8. As we are now in early 2024, there is a whole year of empirical data (epidemiological, vaccine booster uptake) that could be compared to model predictions - currently only up to Aug 2023 - I think this whole-year comparison would be really important to show that model predictions have validity. If model predictions don’t align with 2023 epidemic trends, then the authors should interrogate why this is the case.

9. A more philosophical question about the validity of combining estimates from multiple models using the ensemble approach: Some modelling scenarios are likely to be more plausible than others (more realistic assumptions, better quality input data etc), yet by combining models, do the authors assume that all model scenarios are given equal weight? I think readers will want to see much greater information on the range of estimates obtained from each individual model to allow judgement over whether the ensemble output estimates are plausible. Again, this could be in the form a a table/figure in the main results with 1-2 paragraphs added to describe. (

Comments from the Editorial Team:

GENERAL: 

Please include line numbers in your revised manuscript, ideally not starting from 1 with each new page.

Please provide 95% CIs and p values for all results where appropriate (including the abstract), check and amend throughout. We suggest reporting statistical information in the following format: ‘x’; (95% CI [‘y’,’ z’] p value) and use commas as opposed to hyphens (as these can be confused with negative values) to separate upper and lower bounds. 

For p values, please report as p<0.001 and where higher as 'p=0.002'. Please add the statistical method used to your method section. We also invite you to report p values to consistently to the third decimal digit - thousandths

STUDY DESIGN: 

Of all authors who submit a modelling study we ask for inclusion of specific items, derived from Geoffrey P Garnett, Simon Cousens, Timothy B Hallett, Richard Steketee, Neff Walker. Mathematical models in the evaluation of health programmes. (2011) Lancet DOI:10.1016/S0140-6736(10)61505-X.

Please ensure all the items listed below are included with your manuscript. Please review the list below and confirm/revise as necessary:

i) Please provide a diagram that shows the model structure, including how the disease natural history is represented, the process and determinants of disease acquisition, and how the putative intervention could affect the system.

ii) Please provide a complete list of model parameters, including clear and precise descriptions of each parameter, together with the values or ranges for each, with justification or the primary source cited, and important caveats about the use of these values noted.

iii) Please provide a clear statement about how the model was fitted to the data [including goodness-of-fit measure, the numerical algorithm used, which parameter varied, constraints imposed on parameter values, and starting conditions].

iv) For uncertainty analyses, please state the sources of uncertainties quantified and not quantified [can include parameter, data, and model structure].

v) Please provide sensitivity analyses to identify which parameter values are most important in the model. Uncertainty estimates seek to derive a range of credible results on the basis of an exploration of the range of reasonable parameter values. The choice of method should be presented and justified.

vi) Please discuss the scientific rationale for this choice of model structure and identify points where this choice could influence conclusions drawn. Please also describe the strength of the scientific basis underlying the key model assumptions.

DATA AVAILABILITY:

Thank you for providing all your data on https://github.com/midas-network/covid19-scenario-modelinghub/. However, reviewers had access issues - please double check if correct.

ABSTRACT:

Please structure your abstract using the PLOS Medicine headings: Background, Methods and Findings, Conclusions. Please remove all other subheaders.

Please also note that the abstract should report only the primary scenario outcome or must include ALL tested scenario outcomes 

Abstract Background:

Provide the context of why the study is important. The final sentence should clearly state the study question.

Abstract Methods and Findings:

Please include the study design

In the last sentence of the Abstract Methods and Findings section, please describe the main limitation(s) of the study's methodology.

Abstract Conclusions:

Please begin your Abstract Conclusions with "In this study, we observed ..." or similar, to summarize the main findings from your study, without overstating your conclusions. Please emphasize what is new and address the implications of your study, being careful to avoid assertions of primacy. 

AUTHORS SUMMARY:

Ideally each sub-heading should contain 2-3 single sentence, concise bullet points containing the most salient points from your study.

In the final bullet point of ‘What Do These Findings Mean?’ Please include the main limitations of the study in non-technical language.

ACKNOWLEDGMENTS/ DECLARATIONS

Please remove all statements apart from acknowledgements, author contributions and abbreviations from the end of the main manuscript and include these only in the relevant parts of the manuscript submission form. Funding, competing interest, and data availability will be compiled as metadata.

DISCUSSION:

Please present and organize the Discussion as follows: a short, clear summary of the article's findings; what the study adds to existing research and where and why the results may differ from previous research; strengths and limitations of the study; implications and next steps for research, clinical practice and/or public policy implications; followed by a one-paragraph conclusion.

Please remove all subheadings within your Discussion e.g. Limitations and other considerations.

We also like to ask you to broaden the discussion and reflect on what the results may mean outside the US.

Comments from the reviewers:

Reviewer #1: This is a good piece of evidence about the impact of vaccination on the burden of Covid-19 mortality and hospitalizations. The authors proposed a very interesting approach by putting together different modelling teams to produce their own estimation and then ensemble them together to come out with different potential scenarios.

After reviewing the paper i have the following comments:

- Could the authors develop on the impact on the results regarding the fact that the modelling teams didn't include non-pharmaceutical interventions in their models and that they were free to choose whether to use or not the SARS-CoV-2 seasonality as a factor? In my opinion the seasonality could have a major impact on the two outcomes included in this review (eg. mortality and hospitalizations). Specially if SARS-CoV-2 will be circulating more at the same time as other viruses like influenza or RSV are. The accumulation of patients for multiple circulating seasonal viruses could overwhelm the health facilities and have an impact on mortality. 

- Regarding the target population for the vaccination, even though the authors mention that this is based on the CDC ACIP, in my opinion it would also be important to consider the immunocompromised individuals, specially given the fact that the uptake among this vulnerable group is quite low in the US (Tartof SY, Slezak JM, Puzniak L, et al. Analysis of mRNA COVID-19 Vaccine Uptake Among Immunocompromised Individuals in a Large US Health System. JAMA Netw Open. 2023;6(1):e2251833. doi:10.1001/jamanetworkopen.2022.51833)

- In my opinion is important to highlight in the limitations (although the authors partially do this when they refer to the "strict scenario assumptions") that there are many factor that can influence the number of hospitalizations and deaths apart from the vaccination and the antigenic drifts that were included in the models. I would put emphasis on this for potential readers who are not familiar with the topic.

- The link to the replication codes in GitHub gives a 404 error, meaning that the page doesn't exist. This should be corrected

Reviewer #2: As the public health approach to COVID-19 shifts from emergency response to endemic prevention, understanding both a plausible worst-case scenario and the potential impact of different annual vaccination strategies is important. The authors examine both aspects in their work, leveraging ensemble modeling to provide a look at different scenarios the US might experience with seasonal COVID-19 surges. While the Advisory Committee on Immunization Practices has already provided a blanket recommendation for COVID-19 vaccination in the current 2023-2024 season, the authors' results provide further support for this recommendation. This work may also be useful in supporting future ACIP recommendations in upcoming winter seasons. Overall, the authors have done a fantastic job bo

---

## [Decision Letter · Decision Letter 2]

15 Mar 2024

Dear Dr. Lessler,

Thank you very much for submitting your manuscript "Potential impact of annual vaccination with reformulated COVID-19 vaccines: lessons from the U.S. COVID-19 Scenario Modeling Hub" (PMEDICINE-D-23-03160R2) for consideration at PLOS Medicine. 

[LINK]

Your paper was re-evaluated by a senior editor and discussed among all the editors here. It was also discussed with an academic editor with relevant expertise, and sent to independent reviewers, including a statistical reviewer. The reviews are appended at the bottom of this email and any accompanying reviewer attachments can be seen via the link below

We expect to receive your revised manuscript by Mar 25 2024 11:59PM. Please email me (kjanin@plos.org) if you have any questions or concerns.

If you have any questions in the meantime, please contact me (kjanin@plos.org) or the journal staff on plosmedicine@plos.org. 

We look forward to receiving the revised manuscript by Mar 25 2024 11:59PM. 

Sincerely,

Katrien Janin, PhD

Senior Editor 

PLOS Medicine

plosmedicine.org

Comments and requests from the academic editor and the editorial team:

We thank the authors for the provided explanations and we find the manuscript much improved. We also appreciate you added empirical data up to December 15, 2023. We do have, however, still have a few outstanding issues:

1. We are not convinced by the authors argument that this study is an exercise primarily in projecting the potential future benefits of vaccination. Assumptions for vaccine coverage that feed into projections have not matched the reality of what has happened. Similarly, (albeit to a lesser extent) the modelling input assumptions around seasonality have not borne out. Your main results are reported in terms of potential numbers of deaths and hospitalisations averted under these substantially higher vaccine coverage scenarios, but there is no scenario that reflects the reality of vaccine coverage and epidemic trajectory during 2023. As a result, we think you need to be clearer about this still, and be much more circumspect in the conclusions about the potential impact of COVID on hospitalisations and deaths, given the inputs haven't necessarily reflected reality.

2. We were a bit disappointed you did not take up our suggestion to broaden the discussions to reflect on the possible global implication of your study, and where your study findings sit within a broader context. Given the readership of PLOS Medicine, it is important that you address questions about international implications of your research. Whilst we appreciate it would be inappropriate to make projections for other countries and regions, it would be of value to contrast the modelling here with efforts in other countries/regions.

3. Please address the remaining comment from reviewer #2 (see below)

General editorial comments:

4. Please move the Author Summary. This should be inserted directly after the Abstract. 

5. L 306: “is one of the 306 most robust approaches for estimating COVID-19 … “ please rephrase as this claim is not validated. Suggest: “Despite its limitations, ensembling scenario-based projections from multiple teams is a useful tool for estimating COVID-19’s …” or something similar.

6. . Supplementary Materials: Please note that supplementary materials are not checked and will be posted as supplied by the authors. Therefore, please double check. Please cite your Supporting Information as outlined here: https://journals.plos.org/plosmedicine/s/supporting-information - Please note you may use almost any description as the item name of your supporting information as long as it contains an "S" and number. For example, “S1 Appendix” and “S2 Appendix,” “S1 Table” and “S2 Table", and ensure that all SI materials have a call-out (link) in your main text.

Please feel free to contact me directly at kjanin@plos.org with any questions you may have.

Best wishes,

Katrien

Comments from the reviewers:

Reviewer #1: Thank you to the authors for providing explanations to my comments and to adjust the manuscript accordingly.

I do not have any further comments.

Reviewer #2: The authors have satisfactorily responded to reviewer comments. The addition of Table S1 is particularly helpful and allows the reader to assess where models were similar or different based on discretionary choices. I have only one comment regarding the added information. 

The authors state that they guided teams to implement 65% vaccine effectiveness against symptomatic disease (lines 143-145). In Table S1, why is this described as 65% VE against infection (with only the UVA-EpiHiper team implementing 0% VE against infection and 65% VE against symptomatic disease)? This may seem like a minor quibble, but in the hierarchy of vaccine effects whether or not teams interpreted the guidance as VE against infection, regardless of symptoms, or as VE against symptomatic disease would impact their findings quite a bit. Park, Dushoff, Grenfell, and Weitz do a nice job exploring how varying levels of asymptomatic transmission impacts an epidemic's trajectory (doi:10.1093/pnasnexus/pgad106), which is analogous to how 65% VE against infection would avert a good deal of infections and asymptomatic transmission versus how a 65% VE against symptomatic disease would avert a lower proportion of infections and therefore have a larger infection count for conversion to symptoms/hospitalization. If this is a simple terminology issue (though the description of the UVA-EpiHiper team's choice implies otherwise), I suggest clarifying Table S1. Otherwise this should be acknowledged as a limitation of the work, since at this point teams should not be asked to rerun their models. 

Reviewer #3: The authors have addressed my points

Michael Dewey

[LINK]

---

## [Editor Report · Decision Letter 3]

27 Mar 2024

Dear Dr Lessler, 

On behalf of my colleagues and the Academic Editor, I am pleased to inform you that we have agreed to publish your manuscript "Potential impact of annual vaccination with reformulated COVID-19 vaccines: lessons from the U.S. COVID-19 Scenario Modeling Hub" (PMEDICINE-D-23-03160R3) in PLOS Medicine.

PRESS

Sincerely, 

Katrien G. Janin, PhD 

Senior Editor 

PLOS Medicine

(kjanin@plos.org)